# Applications of Gelatin Methacryloyl (GelMA) Hydrogels in Microfluidic Technique-Assisted Tissue Engineering

**DOI:** 10.3390/molecules25225305

**Published:** 2020-11-13

**Authors:** Taotao Liu, Wenxian Weng, Yuzhuo Zhang, Xiaoting Sun, Huazhe Yang

**Affiliations:** 1Department of Biomedical Engineering, School of Fundamental Sciences, China Medical University, Shenyang 110122, China; 18234508369@163.com (T.L.); 18102489629@163.com (W.W.); zyzyyyzzz@163.com (Y.Z.); 2Department of Chemistry, School of Fundamental Sciences, China Medical University, Shenyang 110122, China; 3Department of Biophysics, School of Fundamental Sciences, China Medical University, Shenyang 110122, China

**Keywords:** GelMA hydrogels, microfluidics, biomedicine

## Abstract

In recent years, the microfluidic technique has been widely used in the field of tissue engineering. Possessing the advantages of large-scale integration and flexible manipulation, microfluidic devices may serve as the production line of building blocks and the microenvironment simulator in tissue engineering. Additionally, in microfluidic technique-assisted tissue engineering, various biomaterials are desired to fabricate the tissue mimicking or repairing structures (i.e., particles, fibers, and scaffolds). Among the materials, gelatin methacrylate (GelMA)-based hydrogels have shown great potential due to their biocompatibility and mechanical tenability. In this work, applications of GelMA hydrogels in microfluidic technique-assisted tissue engineering are reviewed mainly from two viewpoints: Serving as raw materials for microfluidic fabrication of building blocks in tissue engineering and the simulation units in microfluidic chip-based microenvironment-mimicking devices. In addition, challenges and outlooks of the exploration of GelMA hydrogels in tissue engineering applications are proposed.

## 1. Introduction

Tissue engineering has raised considerable attention as a potential alternative to tissue or organ transplantation in the area of biomedical engineering [1,2]. Tissue engineering is a crossing discipline of material science, cell biology, engineering, and life sciences [3,4,5], and it is a rapidly developing field. However, tissue engineering faces various challenges, such as the fact that the amount of tissue regeneration by the tissue engineering technique is much less and can hardly be applied in clinical applications [6]. Other problems may include the difficulties to mimic complex structures and organs as well as the lack of biomaterials with desired mechanical, chemical, and biological properties [7,8]. To meet these challenges, bottom-up assembly methods for producing functional building blocks have emerged [9,10]. The molding method, micro-molding, for instance, is considered as a relatively direct method, which is usually used for manual production of small numbers of samples [11,12]. Although the micro-molding method is known for its simple operation, continuous preparation of hydrogels can hardly be achieved so that the production throughput of this method is limited. In addition, hydrogel structures are easily broken or distorted during the process of demolding [13,14]. It is also difficult to obtain the complex structure of hydrogels with this method [15]. By contrast, the microfluidic technique provides a robust platform to generate tissue engineering building blocks with a series of structures (i.e., particles and fibers) [16] as well as a high throughput, attributed to the continuous production. Combined with the 3-D printing technique, complex scaffolds can be achieved [17,18,19]. In addition, microfluidic chips have shown great potential in the field of tissue or organ mimicking, such as cellular microenvironment simulation [20,21,22]. The microenvironment simulation in microfluidic systems includes the fabrication of cell–cell co-culture/cell–extracellular matrix (ECM) interaction model and organ/tissue-on-a-chip system, etc. [23]. Therefore, microfluidic devices provide a more versatile route to the construction of artificial tissues with higher architectural and cellular complexity.

Despite the progress achieved in microfluidic devic-assisted tissue engineering, the ability to mimic the final tissue orientation still remains a challenge that needs to be addressed [24]. These limitations are in part due to the lack of cell-laden constructs. In this context, the choice of biomaterial is crucial in cell function regulation of tissue engineering. Ideal biomaterials can promote the development of tissue engineering. Hydrogels have been applied in microfluidic systems to meet with various biological requirements due to the extremely hydrophilic polymer network characterized by a high water content and porous structure. Natural hydrogels (i.e., alginate, hyaluronic acid, agarose, chitosan, collagen, fibrin, polyethylene glycol) have raised considerable attention in tissue engineering. However, most hydrogels are limited by the poor mechanical properties and limited cell attachment, GelMA hydrogels, with a series of advantages, such as good biocompatibility (including biosafety and biological functionality), strong cell adhesion, and tunable physicochemical properties, were widely used in the field of tissue engineering. GelMA hydrogels are a type of methacrylamide-modified gelatin that was first reported by Van Den Bulcke et al. [22], which was prepared by reaction of gelatin with methacrylic anhydride (MA) [25,26]. Thus, GelMA was also defined as methacrylated gelatin [22,27,28,29], methacryloyl gelatin [30,31,32,33], and methacrylamide-modified gelatin [23]. However, UV light, which was contained in the fabrication procedure of GelMA, is harmful to the organism, for instance, excessive UV light exposure during cross-linking can reduce the survival rate of cells [34,35,36]. Hence, novel cross-linking methods for GelMA hydrogels have been proposed [37,38,39,40,41,42,43]. Different physical properties and cell response parameters of GelMA hydrogels are of great significance to illuminate the applicability of GelMA hydrogels in different biomedical applications. The physical properties include the compressive modulus [44,45], elastic modulus [46], viscoelasticity [24,47], swelling [36,47], porosity, and degradation performance [48]. The physical and chemical properties of GelMA hydrogels can be adjusted by controlling the synthesis process. For instance, the polymer concentrations are related to the elastic modulus, degradation rate, porosity, and swelling capacity. Photo-crosslinking conditions, such as the UV exposure time, affect the degradation and swelling properties of GelMA hydrogels [49,50]. In addition, the viability, proliferation, differentiation, and spreading of different cells in GelMA hydrogels have been widely studied to demonstrate the biocompatibility of GelMA [51,52,53]. Therefore, GelMA hydrogels are considered as a promising biomaterial in tissue engineering. 

Since 2010, when Nichol et al. [54] demonstrated the use of GelMA in microscale tissue engineering applications based on microfluidics, GelMA hydrogels have been widely applied in tissue engineering assisted by microfluidic systems. In this review, state-of-the-art applications of GelMA hydrogels in tissue engineering assisted by microfluidic devices will be summarized.

## 2. Applications of GelMA Hydrogels as Raw Materials for Tissue Engineering Building Blocks

### 2.1. Raw Materials for Microfibers

Microfibers have been widely used in biomedical applications, especially in the fields of tissue engineering, 3-D cell culture, and cell encapsulation [55,56]. Microfluidic methods for fabricating GelMA hydrogel microfibers are categorized mainly into three types: extrusion, laminar flow, and electrospinning-based methods.

The extrusion method refers to extruding pre-gel solution into a gelator solution by using a syringe needle [56]. Liu et al. [57] reported a novel strategy to fabricate GelMA/alginate microfibers by using the extrusion method. During the process of coaxial extrusion bioprinting, GelMA served as the core (to provide a favorable 3-D microenvironment for cells) and alginate served as the sheath (to support and confine the GelMA hydrogel in the core to allow for subsequent UV cross-linking), as shown in Figure 1a. The strategy could be used to fabricate cell-laden constructs with a tunable 3-D microenvironment. Zhang et al. [58] described a novel method based on microfluidic bioprinting for convenient fabrication of vascularized tissues. In this microfluidic system, a core-sheath microfluidic printhead was used to achieve the smooth bioprinting. A blunt needle was connected to this printhead as the sheath to carry the CaCl_2_ solution. The hybrid bioink composed of alginate and GelMA was delivered through the core of a concentric printhead with a smaller blunt needle inserting into the center of the outer one. A layered microfibrous scaffold formed by co-extrusion of the two flows. The microfiber scaffold was used as a vascular bed by encapsulating endothelial cells in the microfibers. Moreover, the GelMA component allowed for long-term stability of this vascularized tissue, which could be used for in vivo tissue regeneration or in vitro tissue modeling.

In the laminar flow-based method, pre-gel solution and gelator solution would form a coaxial or parallel laminar flow within a microfluidic channel. Shao et al. [61,62] fabricated a type of morphology-controllable GelMA microfiber by using coaxial laminar flow. In this method, the sodium alginate solution and GelMA solution formed a coaxial or parallel laminar flow within a capillary. When the coaxial laminar flow passed through the transparent capillary, elastic GelMA microfibers formed with UV light exposure. This GelMA microfiber with tunable morphological and componential features (core-shell and biphasic structure) could be used for 3-D cell growth. Liu et al. [59] designed a double coaxial laminar flow-based strategy to construct double-layer hollow microfibers that could mimic the native osteon (Figure 1b). A mixed bioink (alginate and GelMA) was used in this work to obtain enhanced mechanical strength and prominent bioactivity. The composite bioink prepolymer was injected into the second and third inlet of the microfluidic device; meanwhile, hyaluronic acid was injected from the first inlet. The multilayer structure of the microfibers was formed during the co-extrusion flowing process. Zuo et al. [63] presented a composite material by combining GelMA with alginate for fiber engineering with double coaxial laminar flow. The microfluidic device was comprised of glass capillaries (one square glass capillary was used to connect two adjacent cylindrical capillaries) and pinheads. Hyaluronic acid was injected into the square glass capillary while the alginate/GelMA composite solution was pumped into adjacent cylindrical capillaries. The microfluidic device enabled the formation of coaxial flow among three fluids owning to the laminar flow effect. Microfibers with a double layer hollow structure were fabricated by using this microfluidic device and these GelMA-based hydrogel fibers were employed to mimic complex tissues. Cheng et al. [64] fabricated bio-active microfibers by employing multiple laminar flow containing GelMA and alginate to create tissue constructs with different functions. In this approach, multiple laminar flows were assembled by aligning scalable three-barrel injection capillaries coaxially within a collection capillary.

Electrospinning is also one of the most widely used methods to generate GelMA hydrogel microfibers. Shi et al. [60] proposed an approach to generate GelMA hydrogel fibers with micro-structured patterns using microfluidic spinning (Figure 1c). The research showed that fibers fabricated at a proper flow rate (100 µL/min) and GelMA concentration (30% *w*/*v*) exhibited uniform and well-arranged grooves on their surfaces. The micro-grooved surfaces could efficiently facilitate cell encapsulation and adhesion. These microfibers could be used as templates for the creation of fiber-shaped tissues or tissue microstructures as well. Chen et al. [65] presented an aligned GelMA hydrogel microfiber scaffold for the repair of a spinal cord injury, which was constructed with GelMA hydrogel and electrospinning technology. Parallel electrode rods were used to fabricate an aligned GelMA fibrous bundle. The porous scaffold consistent with nerve axons was vital to guide cell migration and axon extension. Meanwhile, the scaffold with a high degree of elasticity could resist deformation of a bony spinal canal. In addition, GelMA-based microfibers were used for mimicking cell-laden constructs within the microenvironment. Ebrahimi et al. [66] fabricated GelMA hydrogel microfibers with well-defined surface morphologies using microfluidic spinning technology. Unpatterned GelMA fibers with a smooth surface and micropatterned GelMA fibers with well-defined longitudinal surface microgrooves were generated and excellent cell supportive properties were achieved taking C2C12 myoblasts as study objects. Moreover, the micropatterned GelMA fibers could improve the contractility of engineered muscle constructs. Microfibers with low concentrations of GelMA were more favorable for cellular activity while the poor mechanical properties of low GelMA concentration made the fibers easy to collapse. To ease this shortage, Sun et al. [67] developed a novel template-based microfluidic spinning method to directly assemble magnetic GelMA microfibers and microfibers with low GelMA concentration. GelMA microfibers containing magnetic nanoparticles were synthesized by a microfluidic spinning method. Then, a magnetic micropillar array was designed as a template to mold the microfibers to be a microgrid-like structure (microGC). The method provided a versatile pathway to improve the mechanical properties of fibers made from low-concentration GelMA when used for higher-order cellular assembly. These microGCs also could be used as various 3-D in vitro tissue models in tissue engineering.

### 2.2. Bioink for Complex Structures in Microfluidic Bioprinting Platforms

As a potential and revolutionary technology, bioprinting has been applied in biomedical science. Recent studies have shown that bioprinting is applicable for converting biocompatible materials, cells, and supporting components into complex 3-D functional living tissues. Three-dimensional bioprinting is also applied in regenerative medicine [68]. In addition, the microfluidic bioprinting approach has emerged to print high-fidelity microstructures in the fields of tissue engineering, regenerative medicine, and biosensing [69,70]. In bioprinting, GelMA hydrogels are frequently used as bioink.

Bioink is a key element in the bioprinting platform for tissue engineering based on microfluidic technology [71]. Chen et al. [72] presented a 3-D bioprinted multiscale scaffold integrating the 3-D micro- and macro-environment of native nerve tissue based on a GelMA/chitosan microsphere (GC-MSs) modular bioink (Figure 2). Firstly, a droplet microfluidic system was used to produce the GC-MSs taking GelMA/chitosan solution and mineral oil (with Span 80) as the water phase and oil phase, respectively. With the aid of UV light, the microspheres were solidified. Then, the nerve cells were seeded on the microspheres to prepare the bioink along with GelMA, and last the multiscale composite scaffold was constructed by using the bioink and a 3-D printer. The GC-MSs provided a similar mechanical property with nerve tissue for nerve cell proliferation and differentiation. The round shape of GC-MSs was beneficial for cellular adhesion and proliferation on the surface. Thus, the GC-MS-based 3-D bioprinted composite scaffold was able to protect the nerve cells within a 3-D environment.

Colosi et al. [71] used a mixture of alginate and GelMA as a low-viscosity bioink in a 3-D bioprinter. The study also coupled coaxial needles and a simple microfluidic chip with “Y”-shaped channels to rapidly switch between different bioinks in a fully programmable manner. The microfluidic bioprinting platform with this bioink was conducive to the creation of bio-mimetic tissue models for regenerative medicine and drug discovery applications. Lee et al. [73] developed a GelMA-based 3-D cell-laden scaffold, which could be applied as blood vessels and nerve conduits. This scaffold consisted of hollow pipe struts, which were fabricated by the 3-D cell-printing system supplemented with microfluidic channels, UV treatment system, and low-temperature working plate. During the printing process, the GelMA/dimethyl sulfoxide (DMSO) bioink and the mixture of cell-laden collagen/DMSO bioink were injected in the shell and core regions, respectively, to format the core/shell scaffold. Then, the use of GelMA hydrogels enabled high cell viability.

Knowlton et al. [74] demonstrated a desktop 3-D printer for the rapid prototyping and cost-effective fabrication of a 3-D microfluidic chip. Within this 3-D microfluidic chip, the GelMA prepolymer solution mixed with cells was loaded into a microfluidic channel to couple 3-D cell encapsulation with spatial graphics. Therefore, the microfluidic chip combined with GelMA hydrogels provided a 3-D environment for long-term cell culture and growth. In the latest studies, GelMA hydrogels were used as a part of the printed microchannels. Mansoorifar et al. [75] proposed a re-configurable GelMA hydrogel microfluidic system to mimic human vasculature microcapillary reconfiguration. In this hydrogel microfluidic system, microchannels were printed within GelMA hydrogels. Agarose fibers loaded with iron microparticles were used as bioinks. The microchannels based on GelMA hydrogels provided a microenvironment for cells. The agarose fibers can be used as valves, which could provide the reconfigurability of the system controllable by a permanent magnet.

## 3. Applications of GelMA Hydrogels as Simulation Units in Tissue Engineering

### 3.1. Scaffolds for 3-D Cell Culture

Various hydrogels have been developed for cell culture based on microfluidic devices [76,77,78]. GelMA hydrogels can provide a tunable environment for cells and support their adhesion, growth, and proliferation [79,80,81,82]. In this section, the applications of GelMA hydrogels as scaffolds for 3-D cell culture based on microfluidics will be highlighted.

Takeuchi et al. [83] created a cell culture device with GelMA microstructures. The microstructures were self-assembled inside the microfluidic chip. The fabrication procedure of tubular cell structures using GelMA hydrogel is shown in Figure 3a. In this method, 2-D microstructures were used to assemble the 3-D tube structure. These tubes could be used to mimic the tubular tissue structures made of cells, such as vascular tissue. In addition, the rat cells were seeded on the surface of GelMA microstructures to enhance cell growth. The results of culturing mouse smooth muscle cells on the GelMA microstructures are shown in Figure 3b. Cha et al. [84] presented core-shell GelMA microgels with a highly controllable size by utilizing a flow-focusing microfluidic device. A layer of silica hydrogel was used as a “shell” in order to protect the cells cultured on GelMA microgels. These GelMA microgels could provide a suitable cellular microenvironment that could be used as an in vitro platform to culture cardiac side population (CSP) cells on the surface. These cells on the microgels were able to migrate and spread onto their cell-conductive surrounding. Rahimi et al. [85] introduced a Janus-paper/polydimethylsiloxane (PDMS) platform based on GelMA hydrogels. Janus-paper, with one hydrophobic (polyethylene-coated) face and a hygroscopic one, was bonded to a PDMS substrate with embedded microfluidic channels. Then, the porous network of the hydrophilic side facilitated easy seeding of GelMA hydrogel-encapsulated cells while the function of PDMS microfluidic channels was to deliver nutrients and drugs to the cells. Mahadik et al. [86] described an approach to covalently incorporate stem cell factor (SCF) within a GelMA hydrogel via polyethylene glycol (PEG) tethers. These GelMA hydrogels could be applied to culture primary murine hematopoietic stem cells (HSCs) in vitro.

Moreover, GelMA hydrogels are used for 3-D cell co-culture microfluidic facility, and the most innovative studies are summarized in Table 1. A one-step generation of core-shell GelMA microgels using a droplet microfluidic system was reported by Wang et al. [87]. A core-shell droplet generation unit (a typical flow-focusing microfluidic device) was used to form the laminar flow of the core (methyl cellulose) and the shell (GelMA) flow to generate core-shell microgels. The microgels were used to co-culture liver cells and vascular endothelial cells, and the core acted as the cell-laden scaffolds and the GelMA shell could protect the inner cells in this system. Lee et al. [33] used 10 *w*/*v*% GelMA hydrogel as a semi-permeable physical barrier to control the molecular diffusion in a microfluidic co-culture device. For example, the larger pore size of GelMA resulted in an increased oxygen diffusion rate that promoted cell differentiation. Michigan Cancer Foundation-7 (MCF-7) human breast carcinoma cells and metastatic U87MG human glioblastoma were co-cultured in this device to study photo-thermal therapy efficacy and cancer cell migration. Cui et al. [88] designed and fabricated lobule-like hierarchical micromodules for the assembly of 3-D liver lobule-mimetic constructs consisting of an outer hexagonal GelMA structure containing HUVECs, and inner radial-like PEGDA structure containing human hepatoma cells.

### 3.2. Components for Organs-On-A-Chip

Organs-on-a-chip refers to a bionic system that simulates the in vivo microenvironment of living organs and provides more in vitro models related to the physiology of human organs [92,93]. GelMA hydrogels are widely used in tissue engineering and therefore they are also considered as promising components in organs-on-a-chip.

Nie et al. [92] reported a vessel-on-a-chip system based on GelMA hydrogels. A novel method of preparing a hydrogel microfluidic chip was presented, which combined casting and bonding technology. A vessel-on-a-chip system with vascular function in both physiological and pathological situations was modelled. Gelatin and GelMA served as substrate materials for this microfluidic chip due to their biocompatibility and the property to promote cell functionalization. The manufactural process of the GelMA-based microfluidic vessel-on-a-chip included three steps: casting, demolding, and bonding (Figure 4). This study demonstrated that liable attachment and uniform spreading of human umbilical vein endothelial cells (HUVECs) on the surface of the inner wall of the channels were achieved with a good survival rate and other indicators. In both physiological and pathological situations, microfluidic gelatin-GelMA hydrogel was in favor of realization of vascular function.

Zhang et al. [94] reported a thrombosis-on-a-chip model that included hollow microchannels integrated with confluent endothelial layers. The microfluidic chip could be used to comprehend the significance of cellular interactions and the impact of potential therapeutics to treat thrombosis. In this research, three different models were developed to mimic different aspects of thrombosis and fibrosis. The microchannel covered with endothelium was used as the control group. The other two groups were prepared by covering the microchannels with encapsulated fibroblasts, and encapsulated fibroblasts as well as endothelium, respectively (Figure 5). The results of the blood infusion and perfusion assay revealed that the model that contained hollow microchannels coated with a layer of confluent endothelial embedded in a GelMA hydrogel was able to form thrombi. Migration of fibroblasts into the clot and deposition of collagen type I were further observed, confirming the capacity of the model in mimicking the fibrosis process in vivo. This chip may serve as a potential platform for the fibrosis pathology study.

Li et al. [95] developed a heart-on-a-chip with hydrogel films to mimic the cardiac structure and function. Reduced graphene oxide (rGO)-doped heterogeneous hydrogel film with GelMA hydrogels and polyethylene glycol diacrylate (PEGDA) was utilized to increase the contrast of the structural color as a dark background and to enhance the beating consistency of cardiomyocytes (the conductive rGO was doped in GelMA hydrogel). As shown in Figure 6, the bifurcated film was manufactured via direct UV polymerization (for PEGDA) and mask-defined UV polymerization (for GelMA), as well as an etching process. Then, the film was inserted between the channel and substrate layers of the heart-mimicking chip. On this chip, cardiomyocytes were cultured to test the stimulation effect of isoproterenol, propranolol, and verapamil on the cells. Therefore, this platform was considered to be a promising tool for cardiac pathophysiological study and drug screening [95].

Wang et al. [96] reported a microfluidic tumor progression model based on metastasis-on-a-chip. Decellularized liver matrix (DLM)/GelMA (ratio 2:3) hydrogel was assembled on the chip to serve as a 3-D biomimetic liver microenvironment to culture kidney cancer cells (Caki-1) to mimic the progression of metastatic kidney cancer, as shown in Figure 7. The microfluidic device was constituted by poly(methyl methacrylate) (PMMA) and PDMS. In this system, human-derived hepatocytes (HepLL) were able to continuously produce albumin and urea, which could mimic the biological functions of the liver. 5-Fluorouracil (5-FU)-loaded PLGA-PEG nanoparticles (NPs) were co-cultured with the cells to predict the therapeutic effects and evaluate the efficacy of anticancer drugs. The results demonstrated that the microfluidic tumor progression model could be used to establish 3-D metastatic cancer models in vitro and to rapidly assess anticancer efficiency of certain drugs and optimize the dosage regimes.

Aung et al. [97] described a tumor-on-a-chip (TOC) device created by a novel approach, which could be used as a valid drug-screening platform. In this TOC, GelMA hydrogel structures were integrated into a microfluidic device to package the cancer spheroids along with HUVECs through 3-D photo-patterning. Roberts et al. [98] developed an inclusive on-chip platform, which was capable of maintaining laminar flow through porous bio-synthetic microvessels. This device could deliver and generate a steady perfusion of media containing small-molecule nutrients, drugs, and gases during three-dimensional cell culture where GelMA was used as a scaffold.

In addition, other microfluidic approaches based on GelMA hydrogels were studied in biomedical fields recently [74,83,99].

For instance, Takeuchi et al. [74] developed a procedure for fabricating movable biological cell microstructures using GelMA hydrogels on a microfluidic chip. The microstructures can be used as building blocks for 3-D assembly of in vitro cell structures. The GelMA microstructures were partially dissolved because of their biodegradability after the cells were placed on the microstructures. The degradation of hydrogel may facilitate cell growth because cells tend to grow on the hydrogel surface. In brief, these cell microstructures can be used as a component for assembling a three-dimensional tissue-like cell structure in vitro. Hu et al. [100] reported on-chip acoustic force assembly-assisted GelMA hydrogel arrays to rapidly obtain 3-D ECM mimics were supported multicellular. Each unit encapsulated in the 3-D ECM mimic hydrogel pillars was independent from the other units in the arrays. The GelMA hydrogel arrays can be used in many drug-related applications and could be helpful for guiding personalized treatment for individuals in the clinic. Knowlton et al. [83] presented a novel 3-D-printed microfluidic chip as a controllable 3-D cell culture environment. Cell-laden GelMA hydrogels were demonstrated to provide a suitable environment for cell growth under microfluidic flow conditions. Therefore, cell encapsulation in 3-D hydrogel posts within the microfluidic channels was used to enhance the capabilities of the microfluidic devices. Nan et al. [99] developed a one-chip harvesting of single cell-laden GelMA hydrogels based on a microfluidic system in culture medium. The hydrogels were generated by using an on-chip gelation technique, which included droplet generation under UV light. Then, these GelMA hydrogels flowed into a sorting chamber by active guidance to selectively transfer the cell-laden microgels into the lateral aqueous medium. Moreover, combined with the sorting module, two types of single cells could be respectively encapsulated and collected to realize single-cell analysis.

Generally, GelMA hydrogels provide the organs-on-a-chip system the ability to mimic the microenvironment of living organs and offer more physiologically relevant models of human organs in vitro. GelMA hydrogel-based organs-on-a-chip is also in favor to study the organic biological function in both physiological or pathological situations. Beyond this, GelMA hydrogels can serve as excellent ECM-mimicking materials for encapsulated cells.

## 4. Conclusions and Outlook

With the development of microfluidic technologies, tissue engineering assisted by microfluidic devices has emerged as a promising approach to solve the challenges like complex structures and organs mimicking and construction of cell-laden scaffolds and high-fidelity tissue microstructures. GelMA hydrogels have become an attractive biomaterial category applied in tissue engineering due to their tunable physicochemical properties and good biocompatibility. This work reviewed the recent (mainly after 2015) applications of GelMA hydrogels in tissue engineering assisted by microfluidic devices. The applications mainly include two aspects: (1) GelMA hydrogels serve as the raw materials in tissue engineering assisted by microfluidic devices, such as microfibers and scaffolds based on bioprinting; and (2) GelMA hydrogels serve as the simulation unit, such as organ-on-chip and cell culture systems.

Although microfluidics offers numerous advantages (i.e., miniaturization, accelerated reactions, and automation), the microfluidic technique-assisted GelMA hydrogels still face a series of challenges when applied in the real biomedical field. Firstly, there are many complex structures in real tissues that cannot be simulated through the microfluidic technique at present, and what has been reported are just relatively simple structures like microspheres and microfibers. The inability to create large-scale tissue constructs containing micro-vascularized network channels and the lack of control over long-term cell survival remain unsolved. Secondly, current microfluidic devices also encounter some limitations, such as the feasibility, cost-effectiveness, and large-scale manufacturing. Microfluidics combined with GelMA have not been widely applied for disease modeling, pathophysiological studies, and pharmaceutical research due to the complicated protocols. Thirdly, the dimension of the GelMA hydrogel materials is limited to the micron level. It is difficult to prepare nanomaterials with fine structures via the microfluidic technique even if the 3-D bioprinting method is combined. In addition, the fact that cell death is induced by free radicals, which are produced by the photoinitiators during the cross-linking process, cannot be ignored [45].

In the future work, micro-vascularized networks that can be used to create mesoscale tissue-like constructs may be one of the most essential tasks. Additionally, the mixing of hydrogels with different or complementary properties to form a composite material that may possess enhanced features over a single hydrogel should be encouraged. Furthermore, more innovative and functional microfluidic devices should be set up to meet the demands of various applications in tissue engineering. Beyond the microfluidic technique, the combination with other techniques, such as electro-spraying, nanofluidic devices, or femtosecond laser engraving, to prepare nanoscale GelMA hydrogel-based materials is also an innovative direction. Importantly, exploring mild cross-linking methods that are friendly to living organisms is also a future direction for GelMA preparation.

## Figures and Tables

**Figure 1 molecules-25-05305-f001:**
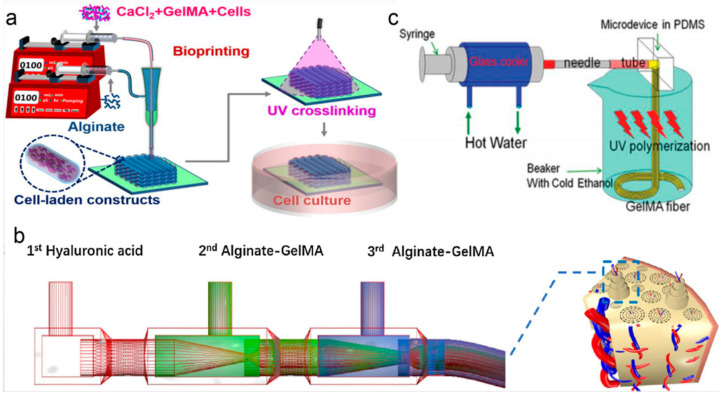
(**a**) Extrusion method: strategy of bioprinting GelMA/alginate core/sheath microfibers into 3-D constructs with tunable microenvironments. Reproduced with permission [57]. (**b**) Laminar flow-based method: Capillary microfluidic device used for biomimetically constructing osteon-like double-layer hollow microfiber with the novel composite bioink. Reproduced with permission [59]. (**c**) Electrospinning method: The microdevice to generate microstructured GelMA fibers. Reproduced with permission [60].

**Figure 2 molecules-25-05305-f002:**
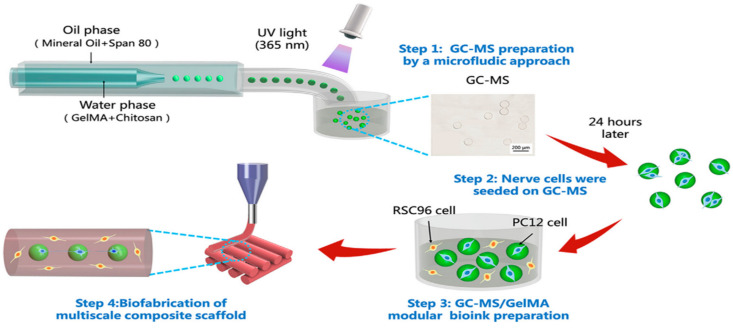
Multiscale composite scaffold preparation based on a gelatin methacryloyl (GelMA)/chitosan microspheres (GC-MSs) modular bioink: GC-MS preparation by a microfluidic approach (step 1), nerve cells seeded on GC-MS (step 2), GC-MS/GelMA modular bioink preparation (step 3), bio-fabrication of 3-D composite scaffold performed by extruding bioink with the 3-D printer (step 4). Reproduced with permission [72].

**Figure 3 molecules-25-05305-f003:**
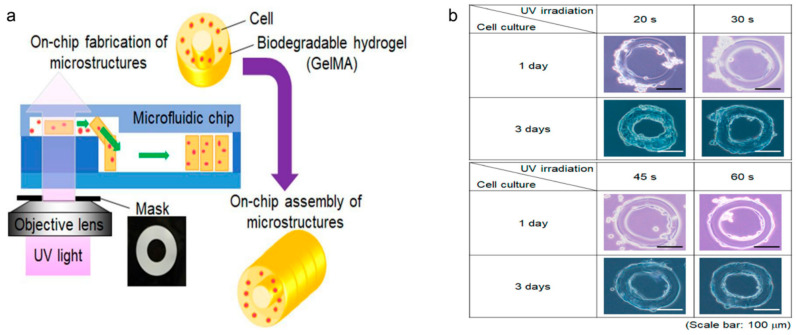
(**a**) Fabrication procedure of tubular microstructures using on-chip fabrication and the assembly of toroidal cell-embedded microstructures. (**b**) Cell culture on GelMA microstructure after 2 days of different UV exposure. Reproduced with permission [83].

**Figure 4 molecules-25-05305-f004:**
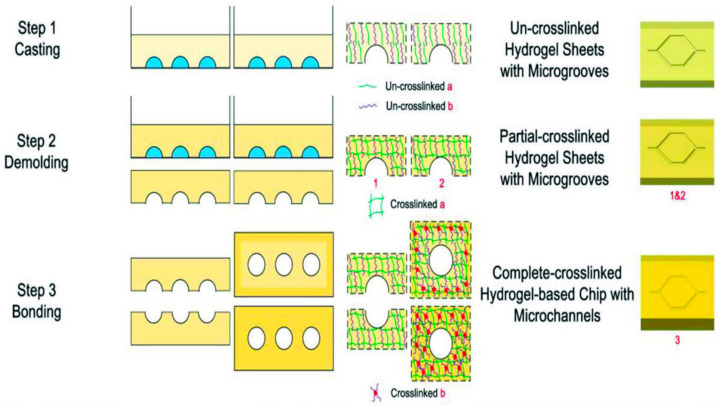
Schematic of the fabrication process for the hydrogel-based chip and its mechanism. Reproduced with permission [92].

**Figure 5 molecules-25-05305-f005:**
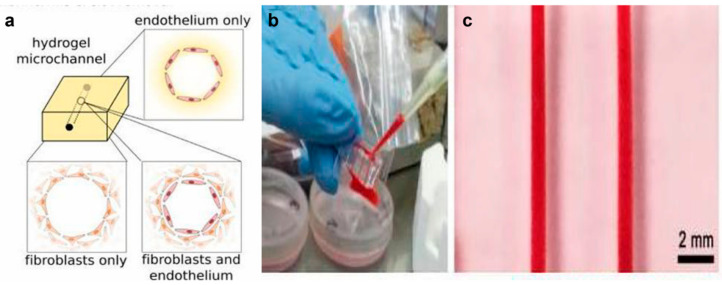
(**a**) To model the different aspects of thrombosis and fibrosis, three different types of models were generated: endothelium covering the microchannel wall with no encapsulated fibroblasts in the matrix (control), encapsulated fibroblasts with no endothelial cells covering the microchannel, and both encapsulated fibroblasts in the matrix and endothelium covering the microchannel. (**b**) Photographs showing the infusion of human whole blood into the endothelialized microchannels and (**c**) the formed thrombosis-on-chip model. Reproduced with permission [94].

**Figure 6 molecules-25-05305-f006:**
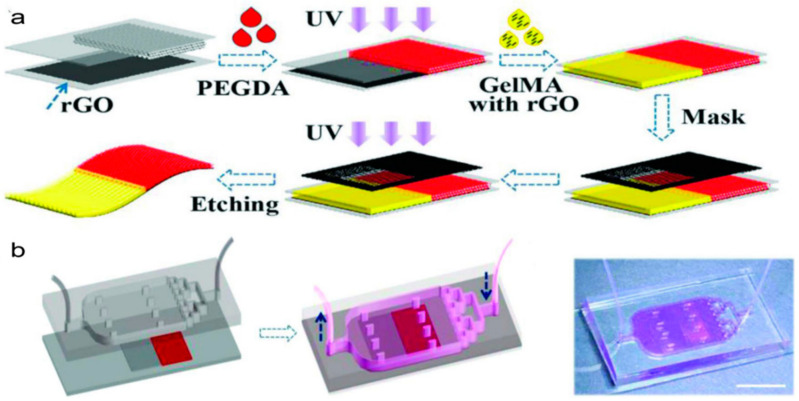
(**a**) Schematic diagram of the generation process of the hydrogel films. (**b**) Schematic and image of the heart-on-a-chip by integrating the rGO hybrid anisotropic structural color film into a bifurcated microfluidic system. Reproduced with permission [95].

**Figure 7 molecules-25-05305-f007:**
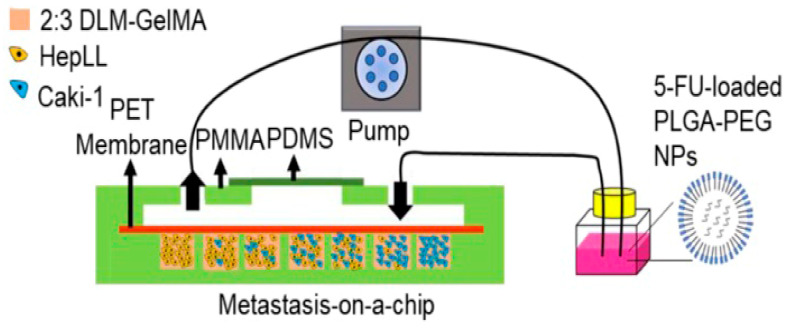
The metastasis-on-a-chip platform. Reproduced with permission [96].

**Table 1 molecules-25-05305-t001:** Summary of the roles of GelMA hydrogels in cell co-culture microfluidic facility.

Hydrogels	Device	Cell Type	Aims and Achievements	Ref.
GelMA	3-D microfluidic device, consisting of five microchambers and four bridge microchannels	Neural stem cells (NSCs) and tumors	Cell co-culture in a 3-D manner	[33]
GelMA, Methyl cellulose, and mineral oil	Droplet flow-focusing microfluidic device	Hepatocytes (HepG2) cells and human umbilical vein endothelial cells (HUVECs)	Core–shell architectures and heterogenous cell cultures	[87]
PEGDA/GelMA	Pressure—assisted hydrodynamic—driven assembly microfluidic chip	HepG2 and HUVECs	Albumin secretion of embedded cells	[88]
GelMA	Digital micromirror device (DMD)-based microfluidic channel	Hepatocytes and fibroblasts	Layered cellular micromodules	[89]
GelMA	Flow focusing microfluidic device	NIH/3T3 fibroblasts	Self-standing microporous environment with an orthogonal void fraction and stiffness	[90]
GO, poly (N-isopropylacrylamide) (pNIPAM) andGelMA	Capillary	HepG2 cells and Hepa1-6 cells	Controllable cell capture and release	[91]

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
