# Peer review of "Applications of Gelatin Methacryloyl (GelMA) Hydrogels in Microfluidic Technique-Assisted Tissue Engineering"

_molecules, 2020, doi:10.3390/molecules25225305_

Round 1
Reviewer 1 Report
This review is interesting as it provides a comprehensive overview of the current protocols using metacrylated gelatin as material for microfluidics based tissue engineering as well as some of the most up to date applications of this technique. As it stands, this review is a good and useful catalog of the state of the art but it does not address the current limitations of existing devices critically, especially in the discussion section. Therefore, it is not easy to envision future avenues of research that could be implemented to overcome these current limitations.
There are a number of minor, yet important issues which essentially concern the formatting of the document.
- References:
- A large number of references do not indicate the year of publication
- I am not sure reference 36 is appropriate as it deals with genipin-crosslinked PEG hydrogels
- I was unable to find reference 41
- I am not sure reference 45, which addresses carboxymethylcellulose and alginate, are appropriate
- In my opinion, reference 53 does not address biocompatibility issues, but only the absence of cytotoxicity
- Paragraph 2.2, line 195-197: I don’t understand the meaning of this sentence “the microfluidic chip….cell growth”
- Table 1: reference 90: please correct “fifibroblasts”
- Line 261: paragraph 3.1 should write 3.2
Author Response
Dear reviewer,
Please see the attachment.
With best regards,
Professor Huazhe Yang
School of Fundamental Sciences
China Medical University, Shenyang 110122, China

Reviewer 2 Report
The authors give a fairy complete overview on gelatin methacrylate based tissue scaffolds fabricated by microfluidic devices. In the reviewer’s opinion no further amendments are necessary.
In my opinion the proposed review is already in a good shape for publication.
It gives a fairly good overview on GelMA for microfluidic based fabrication techniques such as extrusion and electrospinning, microfluidic chip based bioprinting as well as its use to produce hollow GelMA microstructures for cell culture, such as core shell, or whole microfluidic systems towards vascularized organoids.
In addition, different blends and combinations of GelMA with supporting matrices and the respective micromachining methods are presented.
Some minor points to further improve are:
(1) General: The relevance of microfluidic based fabrication methods in contrast to molding methods could explained better and further elaborated.
(2) The introduction might be shortened a bit, fabrication procedure of GelMa can be referenced only as it is not addressed further in the review.
(3) English language can be slightly improved. Among others some typos and grammatic mistakes:
Line 37 “microfluidic chip has shown” should rather be “microfluidic chips have shown”.
Line 39: The sentence “The microenvironment simulation in microfluidic systems includes cell-cell co-culture/cell-extracellular matrix (ECM) interaction model and organ/tissue-on-a-chip system et al. [18].” Is not clear.
Line 50: “first reported by Van Den Bulcke et al. [18]” should be reference 17.
Line 72: “inmicroscale” should be “in microscale”
LINE 241: “Takeuchi, M et al”. “Takeuchi et al.”
Author Response

(The authors gave the same response as above.)

Reviewer 3 Report
This review on the use of GelMA hydrogel in microfluidic technology would be of interest to the microfluidics community. However, I believe the authors can improve the comprehensiveness of the review.
1) Specifically, I suggest including an introductory section on the various hydrogels used in microfluidics and critically compare the advantages/disadvantages of non-GelMA hydrogels to GelMA hydrogels.
2) There is a lack of perspective on where this field is heading. It would be relevant for the authors to provide within the conclusions, or as a separate section, emerging applications of GelMA in microfluidics from the past 3-5 years.
3) Also, critically assess the gaps in current research and really identify where improvements can be made.
Author Response

(The authors gave the same response as above.)

Round 2
Reviewer 3 Report
It is still difficult to understand some parts of the manuscript due to improper grammar. I suggest, at the very least, to have the introduction and abstract reviewed and corrected for grammatical mistakes.